# Reaction of *Cornu aspersum* Immune System against Different *Aelurostrongylus abstrusus* Developmental Stages

**DOI:** 10.3390/pathogens12040542

**Published:** 2023-03-31

**Authors:** Ettore Napoli, Alessandra Sfacteria, Claudia Rifici, Giuseppe Mazzullo, Gabriella Gaglio, Emanuele Brianti

**Affiliations:** Department of Veterinary Sciences, University of Messina—Via G. Palatucci, 98168 Messina, Italy

**Keywords:** *Aelurostrongylus abstrusus*, snail’s immune system, host–parasite interaction, hemocytes, cat lungworm, histology

## Abstract

*Cornu aspersum*, the land snail, is recognized as a suitable intermediate host of *Aelurostrongylus abstrusus*; however, there is little information both on larval development as well as on the intermediate host’s immune system reaction to the parasite. The aim of the study was to assess the histological reaction of *C. aspersum*’s immune system against *A. abstrusus*. Sixty-five snails were provided by a snail farm. Five of them were digested to assess the absence of natural parasitic infections. The remaining sixty were divided into five groups. Three groups of snails were infected with *A. abstrusus* using by-contact or injection methods; one group was injected only with saline solution and one group was left untreated as the control. The snails of group A were sacrificed and digested on study days 2, 10, and 18; snails of the other groups were collected and examined for histopathological analysis on study days 2, 10, and 18. On study day 2, in the infected snails, several free L1s were observed along with the absence of immune system reactions. On day 10, the L2s elicited an intense reaction in the internal layer of the muscular foot. On day 18, all L3s partially encapsulated by the snail’s immune system were observed in the outermost part of the muscular foot, which is near and among the goblet cells. This last finding suggests that L3s could be shed with the snail’s mucus and spread in the environment, representing an alternative route of transmission for this feline lungworm.

## 1. Introduction

Nowadays, the cat nematode *Aelurostrongylus abstrusus* (Strongylida, Angiostrongylidae) has raised increased interest and concern in veterinary medicine due to its spread in cat populations [1,2]. This parasite is diffused worldwide, and in some European countries, it is considered endemic [2]. *Aelurostrongylus abstrusus* features an indirect life cycle [3]; the females, localized in terminal bronchi, alveolar ducts, and pulmonary alveoli, lay eggs that hatch first-stage larvae (L1s) that are shed with host feces [3,4]. In the external environment, *A. abstrusus* L1s can survive for up to 231 days [5] before meeting an intermediate host. Several species of gastropods can be suitable intermediate hosts of *A. abstrusus*. Snails belonging to the genus *Cornu aspersum* (sin. *Helix aspersa*), one of the most common land snails in the world [6], were proven to be an ideal intermediate host in both laboratory and natural conditions [7,8,9,10]. However, the biological relationship and the interactions between *A. abstrusus* and gastropods are poorly investigated, and scarce information on larval development inside the gastropod is available. In particular, the immune reaction of the *C. aspersum* snail against the parasite lacks investigation. It is noteworthy that mollusks, in contrast with mammals, present only innate immune systems in pathogen inactivation [11]; gastropod-specific phagocyte cells, the hemocytes [12], are considered the main effector of the innate immune system [13] that mediates the defense system and replaces the mononuclear phagocytes of mammalian species (i.e., polymorphonuclear neutrophils, monocytes, and macrophages). These cells are also involved in other physiological mechanisms such as intracellular digestion, wound healing, coagulation, and the transport of nutrients [14,15]. It was demonstrated that in some parasitic infections, for instance, those sustained by *Schistosoma haematobium*, the hemocytes recognize and bind to the parasite’s surface and then undergo a cytotoxic activation, resulting in the effective killing of the parasite [16,17,18]; however, in some occasions, this defense system fails and permits parasite development to the infective stage, allowing the perpetuation of the parasite life cycle. The reasons behind the success or failure of the snail’s immune system against parasites are still debated. For parasites featuring an indirect life cycle, such as *A. abstrusus*, the intermediate host could be considered to be a potential filter that modulates infection intensity [19] and parasite dynamics [20]. Therefore, it is easy to understand how the interaction between the intermediate host’s immune system and the parasite plays a crucial role in the perpetuation of the parasite’s biological life cycle. In this sense also for *A. abstrusus*, the study of the hemocytes and the relative activation of the snail immune system is of the greatest importance for the better understanding of processes leading to infection defense as well as those that make the mollusk vulnerable to the parasite.

Therefore, the aims of the present study were (i) to investigate the *C. aspersum* immune reaction against *A. abstrusus* experimental infection and its different developmental stages and (ii) to follow the parasite’s developmental dynamics into the snail’s muscular foot via histology.

## 2. Materials and Methods

### 2.1. Aelurostrongylus abstrusus Larvae Suspension

*Aelurostrongylus abstrusus* L1s were obtained from a naturally infected cat and extracted from the feces by using the Baermann technique [21]. The larvae were concentrated by centrifuging them at 1.678 g for 5 min and identified at species level using morphometrical keys [4,22,23]. Subsequently, L1s were washed two times in sterile saline solution to remove debris and bacteria. They were suspended again and divided into fifty infective doses of 0.1 mL that contained 250 L1s each.

### 2.2. Animals and Housing Conditions 

*Cornu aspersum* snails (n = 65) were provided by a snail farming center that breeds gastropods for human consumption. On study day (SD) 5, the absence of natural nematode infections was assessed by dissecting and digesting 5 snails (see below); the remaining 60 gastropods were randomly divided into 5 groups (namely, group A, B, C, D, and E). Groups A, B, and C were composed of 15 snails each. Group D was composed of ten snails, and group E contained five snails. All groups were housed in the same environmental conditions (+20 ± 3 °C and 16:8 h light:dark cycle) and fasted until the infection day (SD0).

### 2.3. Experimental Protocol

Groups A and B. Each snail was placed individually in a plastic infestation chamber for 48 h. The chamber was composed of six circular wells, each containing a potato slice irrigated with an infective dose containing L1s (n = 250) and covered with a net that was moistened every 8 h [9]. In order to assess the development of each larval stage, the snails of group A were sacrificed and digested in the manner reported by Napoli and collaborators [10] at SD2, SD10, and SD18; the developmental stages of the recovered larvae were assessed according to their morphological and morphometrical features [9].

Group C. Snails were infected through injection of the infective dose (L1s n = 250 in 0.1 mL) in the muscular foot using the method reported in the literature [10]. 

Group D. The snails were injected with 0.1 mL of sterile saline solution in the muscular foot as done for group C [10]. 

Group E. The snails of group E were left untreated and served as the control group.

After the infection procedures, each group was put into dedicated plastic vivaria (45.0 × 30.0 × 10 cm) and kept in the same environmental conditions.

Two snails from groups C and D were sampled 4 h post-infection to investigate the acute impact of the injection method on the immune system. Two snails per each group (B, C, and D) were collected and analyzed 2, 10, and 18 days post-infection (i.e., SD2, SD10, and SD18).

Briefly, each snail was gently euthanized and, after shell removal, were fixed in 10% buffered formalin for 48 h and routinely embedded in paraffin; for each snail, sections 4–5 µm in thickness were stained with hematoxylin and eosin (H&E), Periodic acid-Schiff (PAS), Toluidine blue (TB), and Masson’s Trichrome Stain. Moreover, the natural autofluorescence of the larvae was used to assess larval vitality.

## 3. Results

All of the specimens dissected and digested before the beginning of the study tested negative for the presence of nematodes. No behavioral abnormalities occurred during or shortly after the snail’s infection in any of the study groups, and no one snail died during the study period.

### 3.1. Group A 

The dissection and digestion of the snails of group A revealed the presence of larvae. According to morphological and morphometrical features, larvae were classified as early developing L1s on SD2 (i.e., 393.4 ± 31.5 µm in length and 18.5 ± 3 µm in width), second stage larvae (L2s) on SD10 (i.e., 481.4 ± 45.6 µm in length and 26.5 ± 2.5 µm in width), and L3s on SD18 (i.e., 552.9 ± 41.2 µm in length and 27.8 ± 4.5 µm in width).

### 3.2. Groups B, C, and D 

#### 3.2.1. 4 h and 2 Days Post-Infection (SD2)

In the snails belonging to the C and D groups that were analyzed 4 h post-infection, the wound of the needle was observed in the thickness of the muscular foot (Figure 1a). No differences in histological structure were observed in the muscular foot, and the hemocytes were free in the tissue (Figure 1b) In group C 4 h post-infection and in groups B and C at SD2, several cross-sections of *A. abstrusus* L1s (16.8 ± 2.8 µm in width) were located within the tickles of the snail’s muscular foot (Figure 1c,d); the hemocytes appeared free in the tissue and not yet recruited.

#### 3.2.2. 10 Days Post-Infection (SD10)

In groups B and C on SD10, cross-sections of L2s (22.3 ± 5.6 µm in width) were observed. Some larvae, which were surrounded by a few hemocytes arranged in a thin layer, were apparently intact; conversely, other larvae elicited a severe immune system reaction and were encapsulated by numerous hemocytes organized in a multi-layered pseudo-capsule (Figure 2a), within which the parasites appeared degenerated. 

In detail, the hemocytes at the periphery of the reaction were suggestive of a state of activation due to a vesicular nuclei and foamy cytoplasm, whereas those in close contact with the larvae showed an epithelioid-like appearance, and the larva inside did not present any degeneration (Figure 2b). By contrast, the L2s that provoked a severe immune system reaction showed degeneration phenomena, their cuticles appeared damaged, and it was possible to observe different blebs; these larvae were enveloped by amorphous material, which was likely due to enzymatic activity from the hemocytes (Figure 2c) that aggregated to form a compact layer around the larva. The immune cell aggregation was suggestive of a granuloma-like reaction, as the innermost layer composed by large, rounded cells with clear or foamy cytoplasm, which is indicative of a phagocytic function, whereas the outermost layer was characterized by flattened hemocytes, mimicking a fibroblast capsule (Figure 3b). All larvae, isolated or not by the host’s immune reaction, were in the deepest layer of the snail’s muscular foot (Figure 3a).

By exploiting the autofluorescence produced by both the larvae and the snail’s muscular tissue, it was possible to assess that the larvae degenerated by the snails’ immune system did not produce any fluorescence, whereas the hemocytic pseudo-capsule emitted clear fluorescence (Figure 4a,c).

#### 3.2.3. 18 Days Post-Infection (SD18)

On SD18, all partially encapsulated larvae were seen in the outermost part of the muscular foot (Figure 5a) in both groups B and C. In some snails of group B, it was possible to observe a path due to the larvae’s active migration (Figure 5b). Some larvae were localized among the snail’s goblet cells (Figure 5c); those larvae, observed in correspondence of the larval path, were surrounded by few hemocytes, contrary to the intense immune reaction seen around the second stage larvae at SD10 (Figure 5d). All of the observed larvae, identified as L3s by morphology and dimension (22.3 ± 5.8 µm in width), were enveloped by a capsule composed of degenerated hemocytes; particularly, flattened hemocytes were localized in the part of the capsule oriented towards the innermost layer of the muscular foot, whereas the part of the capsule oriented towards the outermost layer of the muscular foot was characterized by degenerated hemocytes (Figure 5e). In a snail belonging to group C, an L3 was seen breaking the capsule and actively migrating in the muscular foot (Figure 5f). 

## 4. Discussion

In the present study, the reaction of the *C. aspersum* immune system against *A. abstrusus* experimental infection was systematically investigated and described for the first time; the different immune reaction modalities for each developmental stage (i.e., L1s, L2s, and L3s) were highlighted. Moreover, the movement dynamics of *A. abstrusus* larvae within its intermediate host were documented. The hemocytes play a crucial role in parasite recognition, binding, and inactivation [24]; indeed, in this molluscan species, hemocytes are the main efforts of immune defense, as humoral immunity, characterized by opsonins, have the only role to enhance parasite–hemocyte interaction [25]. The lack of natural nematode infections in the snail, as well as the lack of observation of debris or bacterial aggregates digested at SD-5 and along the study period, supports that the immune reactions observed were due exclusively to *A. abstrusus* larvae. Moreover, all snails, before being infected or used as controls and during the study period, were housed in the same environmental conditions. To maintain the different study groups under optimal and standardized conditions, it was mandatory to study their innate immune response because not only pathogens but also other factors (i.e., diet, climate conditions, and stress) could interfere with the snail’s immune response [26,27,28,29]. Of note, no differences were observed between the methods used for the snail’s infection, neither in the immune reaction evoked nor in the development dynamics inside the intermediate host’s muscular foot. Two days post-infection, the L1s were detected randomly in the innermost part of the muscular foot independent of the infection methods employed, suggesting that there is no preferential site within the snail’s foot for larval development. Moreover, the different methods of penetration of the L1s (i.e., percutaneous/oral or iatrogenic by injection) seem not to have influenced immune–response intensity and/or timing, as no differences were observed among different groups. All of the *A. abstrusus* developmental stages were found within the thickness of the snail’s muscular foot. It is notable that most of the developmental larvae (i.e., L2s and L3s) were localized in the muscular foot herein as elsewhere [1,8,30], suggesting that for *A. abstrusus*, as observed for other nematodes such as *Angiostrongylus costaricensis* [31,32], the muscular foot’s massive vascular supply could provide suitable conditions for larval development and shedding. The data obtained in this study showed that the reaction of the naïve snails’ immune system against *A. abstrusus* at their first contact, is slow and progressive. Particularly, in snails the of group C at 4 h post-infection, no hemocytes were observed close to the L1s or in the proximity of the injection site, suggesting that the immune system needs more than two days for complete activation. This finding contrasts with what was observed for *A. costaricensis* in *Omaloxyx sp.* snails, in which initial hemocyte recruitment was observed after 4 h [31]. The delayed recruitment of the immune cells of *C. aspersum* against *A. abstrusus* could be linked to the snail’s poor resistance to the parasites observed for *S. monsoni* in the resistant strains of *Biomphalaria tenegophila*, in which the recruitment of immune cells was observable as early as 1 h post-infection and elicited an intense immune reaction all around the parasite that appeared completely destroyed; conversely, in susceptible strains of *B. tenegophila*, the sporocysts of *S. monsoni* were surrounded by few hemocytes, and the activation of the snail’s immune system was delayed [33]. On SD10, two different scenarios of immune responses to L2s presence were observed: (i) a mild reaction characterized by a few hemocytes of epithelioid-like appearance organized in a pseudo-capsule surrounding an intact larva, and (ii) a severe activation of the immune system characterized by a stronger tissue reaction with numerous hemocytes of histiocytic-like appearance organized in a multi-layer structure surrounding degenerated larvae. It is not clear if these two different scenarios are related (i.e., the latter could be the evolution of the former), or if they must be considered unrelated expressions of the snail’s immune system. Probably, these two modalities of parasite encapsulation could be disposed as two different reactions of the snail’s immune system, where the former is permissive and the second one is restrictive, resulting in immune system aggression and degeneration of the larvae. The gastropod’s hemocyte interaction with different helminths, such as, for instance, trematodes, has been extensively studied [34,35], and it was demonstrated that a stronger immune activation is indicated by faster and more severe encapsulation. It was hypothesized that this feature reflects the degree of host resistance [36,37,38]. As already observed for *S. monsoni*, a fine layer of fibrous cells surrounding an intact parasite are typical of permissive snails’ strain; conversely, the formation of a multi-layer hemocytic capsule lead to the degeneration of the parasite and is a typical expression of the parasite-resistant snail’s immune system [24,33]. The reaction of the snail’s immune system against the L2s seems to be crucial for the progression of the infection. In fact, as evidenced by the data herein presented, an intense activation of the hemocytes that lead to the severe degenerative phenomena of the parasite was observed only against the L2s. By contrast, on SD18, all of the L3s were only partially encapsulated by the snail’s immune system and were observed in the outermost part of the muscular foot near the goblet cells. All of the L3s observed were enveloped by a thin capsule composed of degenerated hemocytes; particularly, flattened hemocytes were localized in the part of the capsule oriented towards the innermost layer of the muscular foot, whereas the part of the capsule oriented towards the outermost layer of the muscular foot was characterized by degenerated hemocytes. All of the observed larvae appeared alive and vital, as demonstrated by their natural autofluorescence and the lack of any presented degenerative phenomena; in this sense, it could be hypothesized that the immune resistance in *C. aspersum* is not strong enough to destroy or isolate all of the larvae. Furthermore, in the present report, some *A. abstrusus* L3s seemed to be able to break the cystic wall and reach the goblet cells layer. Noteworthy, the majority of L3s were located amongst the goblet cells. This last finding suggests that L3s could be shed with the snail’s mucus and spread in the environment, as demonstrated in laboratory conditions for both *A. abstrusus* and *T. brevior* L3s [1]. This route of L3 elimination may represent an alternative route of transmission for feline lungworms. Therefore, stating that this transmission route is also possible in natural conditions and in absence of iatrogenic stimuli, the definitive host does not need to ingest the snail to get infested, but it can contract the parasite via the ingestion of food contaminated by the snail’s mucus containing the infective L3s.

## 5. Conclusions

This study highlighted for the first time the interaction between the *C. aspersum* snail’s immune system and the parasite *A. abstrusus* throughout its development cycle within the gastropod. As previously said, the interaction between the host’s immune system and the parasite plays a crucial role in the perpetuation of the parasite cycle, and even if this study investigated many aspects of this interaction, there are still some points to clarify. Therefore, the next study will be focused on investigating if repeated exposures to the parasite can increase the host’s immune system’s efficacy or make it more permissive. Moreover, and this is also a very important aspect, it is not clear whether the age of the host can influence the immune response. In addition, although the role of fibrinogen-related protein (FREP) in the inactivation of *S. monsoni* in *B. glabrata* has been extensively studied, no information is available about the action of this protein against *A. abstrusus*. 

## Figures and Tables

**Figure 1 pathogens-12-00542-f001:**
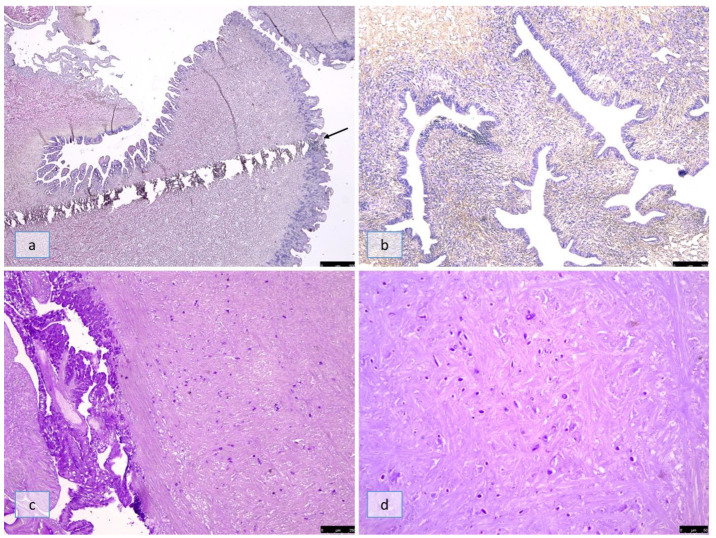
Histological section of the muscular foot of *Cornu aspersum* 4 h post-infection (Groups C and D). (**a**) Path of the needle used for the injection of the infective dose; the hemocytes are free in the tissue (Masson’s trichrome, 5×); (**b**) cross-section of *A. abstrusus* L1s (16.8 ± 2.8). in the snails of group C 4 h post-infection (Masson’s trichrome, 10×); (**c**) group B at SD2; (**d**) group C at SD2. The larvae were located within the tickles of the snail’s muscular foot, and the hemocytes appeared inactivated and free in the tissue (PAS, 5× and 40×).

**Figure 2 pathogens-12-00542-f002:**
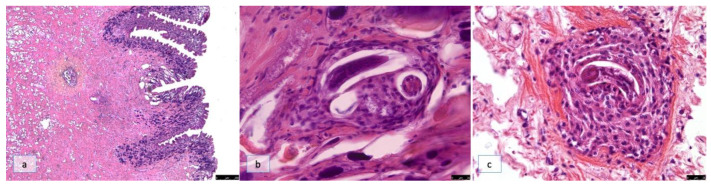
Histological section of the muscular foot of *Cornu aspersum*. (**a**) Groups B and C on SD10 and tissue reactions to L2s (H&E, 10×); (**b**) cross-sections of apparently intact larvae surrounded by activated and epithelioid-like hemocytes (H&E, 40×); (**c**) degenerated L2s enveloped by hemocytes aggregated to form a compact layer around the larva (H&E, 40×).

**Figure 3 pathogens-12-00542-f003:**
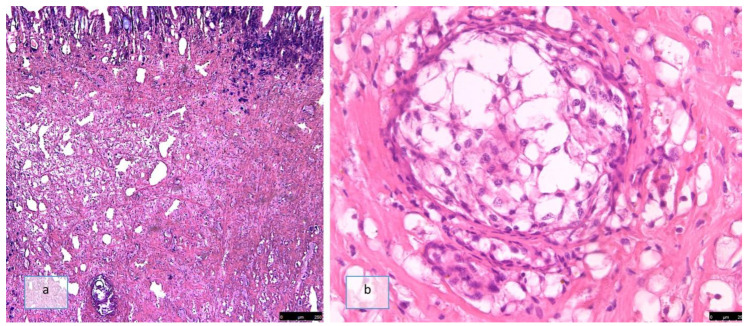
Histological section of the muscular foot of *Cornu aspersum* on SD10. (**a**) All larvae, isolated or not by the host’s immune reaction, were located in the deepest layer of the snail’s muscular foot (H&E, 10×); (**b**) the innermost layer cell of the capsule with large foamy cells and an outermost layer characterized by flattened hemocytes, mimicking a fibroblast capsule (H&E, 63×).

**Figure 4 pathogens-12-00542-f004:**
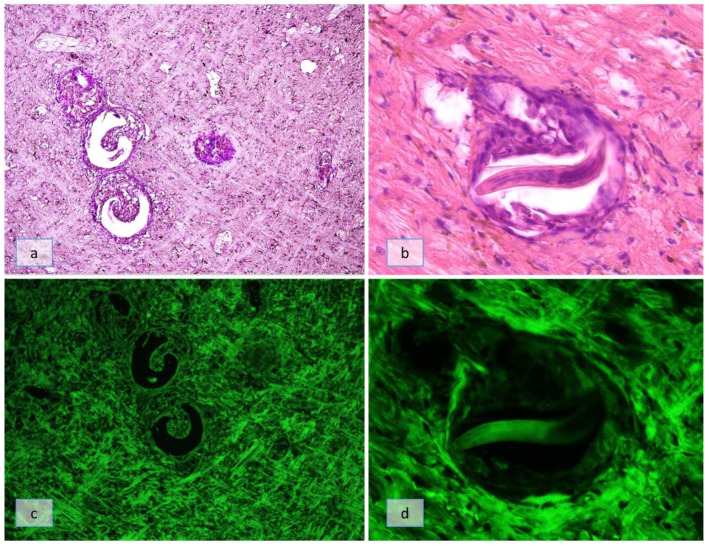
Histological section of the muscular foot of *Cornu aspersum*. (**a**) Dead third stage larvae degenerated by the snails’ immune system (PAS, 20×); (**b**) all apparently vital third larvae (PAS, 40×); (**c**) dead third stage larvae did not produce any fluorescence, whereas the hemocytic pseudo-capsule emitted clear fluorescence (natural fluorescence, 20×); (**d**) vital third stage larva that emitted clear fluorescence (natural fluorescence, 40×).

**Figure 5 pathogens-12-00542-f005:**
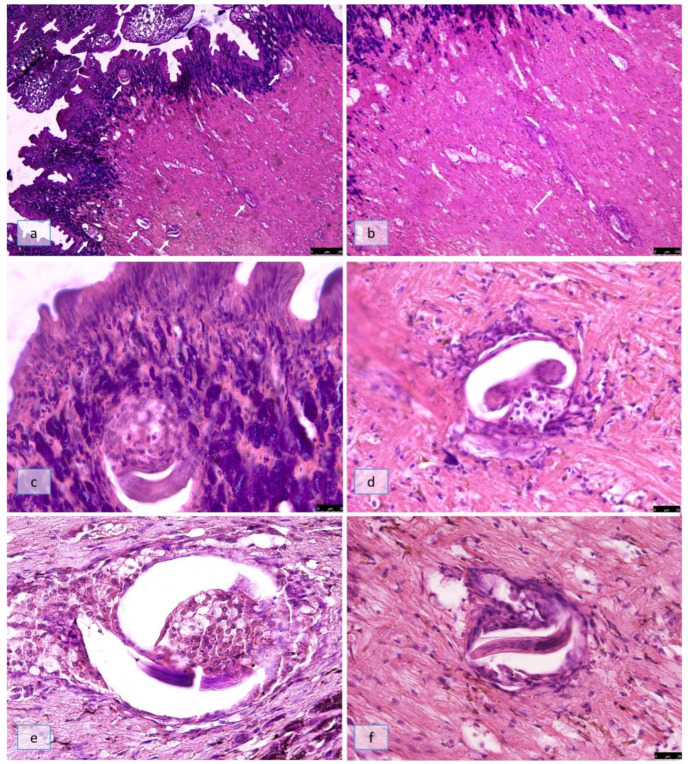
Histological section of *Cornu aspersum* muscular foot on SD18. (**a**) Larvae partially encapsulated by the snail’s immune system observed in the outermost part of the muscular foot (arrows, H&E 5×); (**b**) a path due to larval active migration (arrow, H&E 5×); (**c**) a third stage larvae localized among the snail’s goblet cells (H&E 20×); (**d**) larvae, observed in correspondence of the larval path, surrounded by few hemocytes (H&E 20×); (**e**) third-stage larva enveloped by a capsule composed of degenerated hemocytes (PAS 20×); (**f**) a third-stage larva observed to break the capsule and actively migrate in the muscular foot (H&E 20×).

## Data Availability

Data may be acquired by contacting the corresponding author.

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
