# Peer review of "Reaction of Cornu aspersum Immune System against Different Aelurostrongylus abstrusus Developmental Stages"

_pathogens, 2023, doi:10.3390/pathogens12040542_

Round 1

Reviewer 1 Report

Dear Authors, please find attached my detailed report.
Abstract: please reorganize the abstract, it is quite difficult to follow and read. Remove all the abbreviations and explain better the protocols. 
eg: explain the 5 groups of snails plus the control group ("60 snails divided in 5 groups of 10", not very clear). 
Introduction: appropriate as length and content
Materials and methods: Reorganize this section, it's difficult to read, not all the protocols are properly explained even if the information overall is complete. It took me 3 times to fully understand this part. 
Maybe some tables could help. 
Results: A bit too long, especially because the large number of histological images, nice and clear but try to make it shorter, and difficult to follow and some of them are somehow repeating or unnecessary.
Discussions: logic and to the point.
Conclusions: I would prefer a more concise version of the results moving the last sentences to the discussion section
English should be revised during the proof reading process

Author Response

Reviewer 1

Dear Authors, please find attached my detailed report.

Dear reviewer,

thanks for your precise and informative notes and for reading and appreciating our study. We have revised the paper according to your suggestion and we hope that it is now clearer. Please find the corrections in the track system revised copy.

Q: Abstract: please reorganize the abstract, it is quite difficult to follow and read. Remove all the abbreviations and explain better the protocols.  eg: explain the 5 groups of snails plus the control group ("60 snails divided in 5 groups of 10", not very clear).

A: We apologize, unfortunately we missed that the text final version was quite confusing. The total number of snails was 65 of which 5 used to assess the lack of spontaneous infections. The other 60 were divided in 5 groups: A, B and C of 15 snails each; D and E of 5 snails each.

Q: Introduction: appropriate as length and content

A: Thank you.

Q: Materials and methods: Reorganize this section, it's difficult to read, not all the protocols are properly explained even if the information overall is complete. It took me 3 times to fully understand this part.  Maybe some tables could help.

 A: Materials and methods have been revised according to yours and reviewer 2 suggestion.

Q: Results: A bit too long, especially because the large number of histological images, nice and clear but try to make it shorter, and difficult to follow and some of them are somehow repeating or unnecessary.

A: Results have been revised. One image has been removed, others have been put as insert to make the images more self-explanatory.

Q: Discussions: logic and to the point.

A: Thank you. Discussions have been edited for the language.

Q: Conclusions: I would prefer a more concise version of the results moving the last sentences to the discussion section

A: We have revised and shortened the results in the attempt to make them straightforward.

Q: English should be revised during the proof reading process.

A: A deep revision and editing for language have been performed.

Reviewer 2 Report

The invasion mechanism of parasites and the immune response of the body have attracted much attention. Because snails are a staple of the dinner table, and with the trend of keeping exotic pets on the rise, the immune response between parasites and snails is worthy of attention. However, before publication, there are some places need to modify.

1.      Abstract The abstract content mainly describes groups and results, and the framework of the full text is not stated, which is not easy to understand.

Line 12 Is the sample size 60 or 50?

2.      Introduction There is no analysis of the current situation, and the recently published literature needs to be added.

3.      Materials and Methods There is no structure in the current writing format, suggesting subheadings.

4.      Results Can the immune response also be reflected from other physiological states of snails? The results are only histopathological sections, which cannot fully demonstrate the immune response between parasite and host.

5.      Discussion The language is not fluent and the writing is not standard. Please pay attention to the format of the biological name.

Author Response

Reviewer 2 

The invasion mechanism of parasites and the immune response of the body have attracted much attention. Because snails are a staple of the dinner table, and with the trend of keeping exotic pets on the rise, the immune response between parasites and snails is worthy of attention. However, before publication, there are some places need to modify.

Dear reviewer,

thanks for your precise and informative notes and for reading and appreciating our study. We have revised the paper according to your suggestion and we hope that it is now clearer. Please find the corrections in the track system revised copy.

Q: Abstract The abstract content mainly describes groups and results, and the framework of the full text is not stated, which is not easy to understand. Line 12 Is the sample size 60 or 50? The abstract was deeply revised according to your suggestion.

A: We apologize if the text final version was quite confusing. The total number of snails was 65 of which 5 used to assess the lack of spontaneous infections. The other 60 were divided in 5 groups: A, B and C of 15 snails each; D was composed of 10 snails; E of 5 snails.

Q: Introduction There is no analysis of the current situation, and the recently published literature needs to be added.

A: Done

Q: Materials and Methods There is no structure in the current writing format, suggesting subheadings.

A: Done

Q: Results Can the immune response also be reflected from other physiological states of snails? The results are only histopathological sections, which cannot fully demonstrate the immune response between parasite and host.

A: All the groups were housed at the same environmental condition (+20 ±3 °C and 16:8 h light: dark cycle) and two groups (D and E) were used as controls. Differences in the hemocyte reaction were striking between infected snails and control ones. The purpose of the paper was of describing, at first, the cellular reaction against the developmental stages of the parasite and the localization of L3 larvae in the snail body.

Q: Discussion The language is not fluent and the writing is not standard. Please pay attention to the format of the biological name.

A: A deep revision and editing for language have been performed.

Round 2

Reviewer 2 Report

After revision, the article is more readable. The experimental methods and results are clearly described. Another point that needs to be noted most is that the content of this study is relatively simple, only focus on the histological reaction. The experimental design is not complete enough. Could the record of the results be more comprehensive? Whether snails have other indications of inflammatory, mechanized immune processes.

Author Response

Dear Reviewer, many thanks for your suggestion. However, in this research our main goal was to describe the histological reactions observed in the snail during the developmental stages of A. abstrusus. Further researchers and our and ongoing work are focusing on other indicators of innate inflammation both humoral and cellular. 

Thank you once again for your time and your revision work. 

Kind regards